# Investigation on Wire Electrochemical Discharge Micro-Machining

**DOI:** 10.3390/mi14081505

**Published:** 2023-07-27

**Authors:** Weijing Kong, Ziyu Liu, Rudong Zhang, Yongbin Zeng

**Affiliations:** College of Mechanical and Electrical Engineering, Nanjing University of Aeronautics and Astronautics, Nanjing 210016, China; kongwj@nuaa.edu.cn (W.K.); lzynuaa@163.com (Z.L.); zrd4615@163.com (R.Z.)

**Keywords:** micro hybrid machining, wire electrochemical discharge machining, electrochemical, discharge

## Abstract

With the development of MEMS, the machining demand and requirements for difficult-to-machine metal micro parts are getting higher. Microelectric discharge machining is an effective method to process difficult-to-machine metals. However, the recast layer caused by high temperatures in microelectric discharge machining affects the properties of machined materials. Here, we propose the wire electrochemical discharge micro-machining (WECDMM) and develop a new electrolyte system, which removes the recast layer. In this study, the mechanism of WECDMM was elucidated. The electrolyte was optimized through a comparison experiment, and NaNO_3_-glycol solution was determined as the best electrolyte. The influences of key process parameters including the conductivity of the electrolyte, pulse voltage, pulse-on time and wire feed rate were analyzed on the slit width, standard deviation, the radius of fillet at the entrance of the slit and roughness. Typical microstructures were machined, which verified the machining ability of WECDMM.

## 1. Introduction

Micro-electronic mechanical system (MEMS) has the advantages of small size, high integration, high reliability, and low energy consumption. MEMS is widely used in various fields such as aerospace, precision instruments, micro-robots, and microsensors. With the development of MEMS, the machining demand and requirements for difficult-to-machine metal micro parts are getting higher. It needs not only to have a sufficiently small feature size, but also has characteristics such as high machining accuracy, high surface quality, and large aspect ratio.

Micro-cutting is an important branch of micromachining technology, which generally does not require customized specialized tools, has a short processing cycle, and high flexibility. At present, micro-cutting includes ultra-short pulsed laser cutting, micro water jet cutting, microwire electrochemical machining, microwire electric discharge machining, and so on. Ultra-short pulsed laser realized micromachining at the micron and submicron levels due to its ultra-narrow pulse width and extremely high instantaneous energy density [1]. However, the heat-affected zone is inevitably generated. Micro water jet cutting uses a highly concentrated water jet to remove materials of workpieces, which has advantages such as green environmental protection and no heat-affected zone [2]. However, there may be deformation caused by residual stress. Both micro-wire electric discharge machining (MWEDM) and micro-wire electrochemical machining (MWECM) are non-contact machining technologies, which are very suitable for micromachining of difficult-to-machine metals due to the absence of cutting forces during machining. MWEDM removes workpiece materials through the high temperature generated by discharge, which has high machining accuracy and efficiency [3]. Discharge has an extremely high instantaneous energy and can corrode any metal material. The explosive force generated by sparks is also conducive to expel products from the machining gap. However, the high-temperature effect of discharge produces a recast layer on the machined surface, reducing the quality of the machined surface. MWECM is based on the principle of electrochemical anodic dissolution to remove the workpiece material [4]. There is no recast layer generated during the processing. However, the machining efficiency and precision lag behind MWEDM. Especially, when machining structures with large aspect ratios, the mass transfer in narrow and long micro machining gaps are severely restricted, which greatly affects the stability and efficiency of processing.

Wire electrochemical discharge machining (WECDM) is an effective method to process non-conductive materials such as glass and quartz. The gas film generated by the electrochemical reaction isolates the electrode and electrolyte, providing conditions for discharge. The workpiece near the electrode is removed by discharge. Oza et al. [5] used zinc-coated brass wire to machine quartz, reducing the wire breakage during WECDM. Rattan et al. [6,7,8] used a magnetic field to improve the material removal rate and surface roughness in traveling wire electro-chemical spark machining. Kuo et al. [9] used titrated electrolyte flow for WECDM of quartz glass, reducing electrolyte consumption. Kuo et al. [10] enhanced the surface quality by adding SiC powder in titrated electrolytes during WECDM. Liu et al. [11,12] proposed the WECDM using a rotating helical tool for processing ultra-clear glass. Chen et al. [13] enhanced the ability of WECDM for high aspect ratio glass structures by ultrasonic vibration. Kumar et al. [14] studied the investigation of the effect of the governing process parameters on material removal rate and surface roughness during the WECDM of quartz. Yang et al. [15] reduced unstable discharge phenomena and discharge heat generation by ultrasonic-assisted WECDM with continuous electrolyte flow.

With the deepening of research, the processing objects have gradually expanded to metal, metal matrix composite and other conductive materials. Liu et al. [16] studied the behavior of WECDM of Al6061/Al_2_O_3_ MMC. ElHofy et al. [17] proposed the wire electrochemical arc machining and discussed the effects of the mode of electrolyte flushing wire erosion and machining speed on metal removal rate and accuracy. Shamim et al. [18] used a water mist formed by mixing air and aqueous NaOH solution as a medium for processing Al6063/SiC/10p MMC. Wu et al. [19] used deionized water as the medium to conduct electric discharge electrochemical machining experiments on 2 mm-thick 304 stainless steel. The results indicate that surface roughness was reduced by 63% compared to wire electrical discharge machining. Zhang et al. [20] used an emulsion with a certain conductivity (12.88 mS/cm) as the medium and adjusted the ratio of discharge and electrolysis by controlling the no-load rate to achieve no recasting layer processing. Kong et al. [21] created conditions for hybrid machining using low-conductivity NaNO_3_–glycol solution and realized the machining of a 30-mm-thick workpiece without a recast layer. So far, no one has conducted research on microwire electrochemical discharge machining.

In this study, wire electrochemical discharge micro-machining (WECDMM) is proposed. Coexistence of discharge and electrochemical dissolution for material removal using low conductivity salt solution. A rapid removal of frontal workpiece material through discharge. The surface of the slit sidewall is ultimately formed by electrochemical dissolution, which improves the quality of the machined surface. Microstructures such as micro spiral structures and micro pentagram structures are manufactured with optimized parameters.

## 2. Machining Principle of WECDMM

The prerequisite for implementing WECDMM is that the machining medium has not only a certain degree of conductivity to undergo electrochemical reactions but also meets the requirements of discharge under certain conditions. Zhang et al. [22] and Kong et al. [21] have shown that electrochemical discharge machining of metal materials can be achieved in low conductivity salt solutions. The proposed WECDMM uses the low conductivity salt solution as the machining medium. The schematic diagram of WECDMM is shown in Figure 1. Low-speed unidirectional traveling of wire electrodes is beneficial for achieving high-quality processing and avoiding defects such as reverse stripe. The maximum gap for discharge is *d_f_*. The distance between the wire electrode and the workpiece is *d*. The machining area can be divided into hybrid machining area and electrochemical machining area.

Stage 1. *d ≤ d_f_*. In the hybrid machining area, discharge and electrochemical reactions concurrently exist, and discharge is dominant. The workpiece material is rapidly removed by discharges, achieving high machining efficiency. However, an electrochemical reaction is insufficient to remove the recast layer formed by discharge.

Stage 2. *d > d_f_*. As the machining gap is greater than the discharge gap, discharge cannot occur. The recast layer on the surface of the slit sidewall is removed by electrochemical reaction. The uneven surface formed by discharge is leveled and the surface quality is improved.

## 3. Experimental Apparatus

In this study, the electrode adopts tungsten wire with a diameter of 50 μm, combined with high-precision CNC motion, to achieve precise machining of micrometal parts. An experimental system was designed, as shown in Figure 2. The experimental system includes a unidirectional wire traveling system, an X-Y-Z motion stage for high-precision positioning, an electrolyte circulation system, a pulse power supply, and a vibration isolation platform. The workpiece was connected to the high-precision X-Y-Z motion stage through a fixture to achieve relative feed motion with the wire electrode. Part of the electrodes and workpiece were immersed in the electrolyte. A filter device removed insoluble machining products. A high magnification Charge Coupled Device (CCD) camera was used to record the machining phenomena. An oscilloscope was used to record the voltage and current waveform during machining.

The workpiece was a 9Cr18Mo thin plate (20 mm × 25 mm × 1 mm). The machining conditions are listed in Table 1. A digital microscopy system (VHX-6000, KEYENCE, Osaka, Japan) was used to measure the width, standard deviation and chamfering of the slit. An atomic force microscope (Dimension Edge SPM; Bruker, Karlsruhe, Germany) was used to measure the roughness of the slit sidewall.

## 4. Results and Discussions

### 4.1. Selection of Electrolyte

The electrolyte must not only achieve an electrochemical reaction but also meet the requirements of discharge. Both low conductivity salt–aqueous solution [6] and low conductivity salt glycol solution [5] realized electrochemical discharge machining of metallic material. This study used NaCl and NaNO_3_ as alternative salts deionized and glycol as alternative solvents. NaCl–aqueous solution, NaNO_3_–aqueous solution, NaCl–glycol solution and NaNO_3_–glycol solution were obtained by combining salt and solvent.

The final machined surface is formed by electrochemical machining in WECDMM. The electrochemical dissolution characteristics of different electrolytes were studied. The typical curve in different electrolytes was recorded on an electrochemical workstation (ZENNIUM pro, ZAHNER, Kronach, Germany) to characterize the polarization of 9Cr18Mo, as shown in Figure 3. Saturated calomel electrode (SCE) was used as a reference electrode. As can be seen, the Active dissolution potential and transpassive potential of the salt–glycol solution are lower than that of the salt–aqueous solution. The transpassive current density of the salt–glycol solution is higher than that of the salt–aqueous solution. Transpassive potential indicates that the passivation regime is more likely to occur in the salt–aqueous solution than the salt–glycol solution. Transpassive current density indicates that the corrosion rate of 9Cr18Mo in the salt–glycol solution is higher than the salt–aqueous solution. Similarly, the transpassive current density of the NaNO_3_–aqueous solution and the NaNO_3_–glycol solution is lower than that of the NaCl–aqueous solution and the NaCl–glycol solution. It proves that NaCl is more likely to corrode 9Cr18Mo than NaNO_3_. In addition, a second activation peak appears in the polarization curves of the NaNO_3_–aqueous solution and the NaNO_3_–glycol solution. The workpiece processing area was polished within the potential range of the second activation peak.

The conductivity of the salt solutions is controlled to 100 μS/cm. Other parameters are as follows: voltage of 45 V, pulse on-time of 5 μs, pulse off-time of 12 μs, wire traveling speed of 10 mm/s and feed rate of 2 μm/s. The morphology of the micro slits machined by WECDMM with different salt solutions is shown in Figure 4. The precision of the micro slit machined in the salt solution is poor. It may be related to lots of short circuits during the machining. A passivation regime is likely to occur in the salt solution. The non-conductive passivation film caused frequent short circuits during the machining. Products were welded on the machined surface using the heat generated by a short circuit. As a comparison, the micro slit machined with salt glycol solution has higher accuracy. Spurious corrosion occurred in the unmachined area in NaCl–aqueous solution, and NaCl–glycol solution. This is because Cl^−^ has a stronger activation effect, making the anode more susceptible to corrosion. In summary, NaNO_3_–glycol solution was selected as the electrolyte.

The conductivity of the electrolyte mainly affects the electrochemical reaction in WECDMM. The higher the conductivity of the electrolyte, the more intense the electrochemical reaction. The conductivity of the electrolyte is set to 10 μS/cm, 30 μS/cm, 50 μS/cm, 70 μS/cm and 90 μS/cm. The influence of electrolyte conductivity on the slit width, standard deviation, the radius of the chamfering, and surface roughness is shown in Figure 5. Figure 6 shows the micro-morphology of slit sidewall machined using NaNO_3_–glycol solution with various conductivities. When the conductivity of the electrolyte is increased, the electrolysis reaction is significantly enhanced, while the discharge is less affected. As a result, with the conductivity of the electrolyte increasing, the slit width and the radius of the chamfering linearly increase, while the standard deviation changes little. Moreover, with the enhancement of electrochemical reaction, the defects on the machined surface such as the discharge pit and recast layer are optimized; the surface roughness decreases. However, excessive chamfering radius affects the machining accuracy. The conductivity of the electrolyte needs to be selected according to different processing requirements.

### 4.2. Phenomenon of WECDMM

In order to clarify the discharge characteristics in low conductivity NaNO_3_–glycol solution, the machining phenomenon of WECDMM was observed by a CCD camera, as shown in Figure 7. Figure 7a shows the picture recorded without machining. When pulse voltage is applied to the anode and cathode, hydrogen bubbles form on the surface of the electrode. Bubbles diffuse and eventually fill the machining gap, as shown in Figure 7b. As the relative feed between the workpiece and electrode, the machining gap gets smaller until a discharge occurs. The discharge occurs in a numerous bubbles environment, as shown in Figure 7c. Bubbles are more likely to be broken down by discharge because the dielectric strength of gas is lower than that of liquid. During the process of discharge formation, the machining gap is filled with bubbles. Thus, the medium of discharge breakdown is a bubble in an environment of numerous bubbles. The explosive force generated by discharge causes nearby bubbles to rupture or move away from the gap. If the discharge occurs densely, there is no time for bubbles to fill the gap. The discharge occurs in an environment with few bubbles, as shown in Figure 7d. Compared with Figure 7c, there are fewer bubbles in the machining gap in Figure 7d, and the contour of the electrode wire can be seen clearly. Thus, the medium of discharge breakdown is low conductivity NaNO_3_–glycol solution in an environment of few bubbles.

Voltage and current waveform of WECDMM as shown in Figure 8. As can be seen, the current of ECM is about 40 mA, the peak current of discharge is about 1 A, and the maintaining voltage of discharge is about 22 V. Figure 8a shows the waveform of sparse discharge. There is sufficient time for the bubbles to regenerate and arrange before the next discharge, thus, the discharge occurs in an environment of numerous bubbles. Figure 8b shows the waveform of dense discharge. The distribution of bubbles is destroyed by the explosive force of dense discharge. Thus, the medium of discharge breakdown is the solution.

### 4.3. Effect of Voltage, Pulse On-Time and Wire Travelling Speed

A detailed study was conducted to determine the influence of machining parameters on characteristics. Four parameters such as pulse voltage, pulse on time and wire traveling speed were studied. Each group of experiments was set with five levels.

#### 4.3.1. Effect of Pulse Voltage

Pulse voltage is a key parameter that reflects the output energy of a pulse power supply. When other parameters are constant, the higher the voltage, the greater the machining energy. Pulse voltage is set to 35 V, 40 V, 45 V, 50 V and 55 V. The machining used a NaNO_3_–glycol solution with a conductivity of 30 μS/cm, pulse on-time of 5 μs, pulse off-time of 12 μs, wire traveling speed of 10 mm/s and feed rate of 2 μm/s.

The influence of voltage on the slit width and standard deviation, the radius of the chamfering, and surface roughness are shown in Figure 9. As the voltage increases, the slit width and the radius of the chamfering linearly increase, while the standard deviation changes with a fluctuation range of 0.2 μm. It is because as the voltage increases, the intensity of discharge and electrochemical reaction increases, the ability to remove materials increases, and the width of the gap increases. The radius of the chamfering increases with the increase in electrochemical reaction intensity. The roughness of the machined surface increases with the increase in voltage. It is because the pits produced by discharge are larger and deeper when the voltage is greater. Meanwhile, an electrochemical reaction is insufficient to dissolve rough surfaces formed by discharge.

#### 4.3.2. Effect of Pulse On-Time

Pulse on-time is the effective machining time within a pulse cycle. Pulse on-time is set to 1 μs, 3 μs, 5 μs, 7 μs and 9 μs. The corresponding duty cycles are 9.1%, 23.1%, 33.3%, 41.2%, and 47.4%. The machine used a NaNO_3_–glycol solution with a conductivity of 30 μS/cm, a voltage of 45 V, a pulse off-time of 12 μs, a wire traveling speed of 10 mm/s and a feed rate of 2 μm/s.

The influence of pulse on-time on the slit width and standard deviation, the radius of the chamfering, and surface roughness are shown in Figure 10. With the increase in pulse on-time, slit width, standard deviation and radius of fillet significantly increase. It is because the peak current of discharge increases with the increase in pulse on-time, enhancing the ability of discharge. Moreover, when the pulse off-time and total machining time remain unchanged, with the increase in effective machining time within a pulse cycle, the effective machining time of the entire machining time increases. As the material removal rate increases, slit width increases. The radius of the fillet also increases with the increase in the entire ECM time increase. As the material removal rate increases, the products generated by WECDMM increase, the possibility of short circuits increases, and the standard deviation increases. Due to the increase in peak current of discharge, the discharge pits get larger and deeper, resulting in an increase in surface roughness. The time of electrochemical reaction also increases. However, its strength is limited and it cannot dissolve the rough surface caused by the increase in pulse on-time. 

#### 4.3.3. Effect of Wire Travelling Speed

The wire traveling speed affects the mass transfer in the machining gap. In micromachining, mass transfer within a narrow machining gap has a significant impact on machining stability and quality. Wire traveling speed is set to 5 mm/s, 10 mm/s, 15 mm/s, 20 mm/s and 25 mm/s. Other machining parameters are as follows: NaNO_3_–glycol solution with conductivity of 30 μS/cm, voltage of 45 V, pulse on-time of 5 μs, pulse off-time of 12 μs and feed rate of 2 μm/s.

The influence of wire traveling speed on the slit width, standard deviation, the radius of the chamfering, and surface roughness is shown in Figure 11. With the increase in wire traveling speed, slit width increases slightly, the standard deviation decreases to 1.5 μm and remains stable. The radius of the chamfering and surface roughness shows a decreasing trend. When the wire traveling speed is low (5 mm/s), the products in the machining gap cannot be removed smoothly. Especially, the products are prone to accumulate at the exit. Due to an increase in poor machining conditions such as short circuits, electrochemical machining is limited. Thus, the slit width is small and the standard deviation is large, while the radius of the chamfering and surface roughness are also large. When the wire traveling speed is increased to 15 mm/s, there is a significant improvement in the mass transfer. When the wire traveling speed is increased further, slit width, standard deviation, the radius of the chamfering, and surface roughness are not significantly improved. But the cost has increased. Therefore, the range of wire traveling speed from 10 mm/s to 15 mm/s is more suitable.

### 4.4. Recast Layer in WECDMM

The presence of a recast layer leads to a decrease in the quality of the machined surface. A recast layer is inevitable in MWEDM, as shown in Figure 12. Electrochemical reaction dissolves the recast layer during the process of removing the material by discharge in WECDMM. Metallographic analysis was conducted on the cross-section and longitudinal section of the slit. In order to demonstrate the ability of WECDMM, the slit that was machined using NaNO_3_–glycol solution with conductivity of 10 μS/cm was selected for analysis. The other parameters are as follows: voltage of 45 V, pulse on-time of 5 μs, pulse off-time of 12 μs, wire traveling speed of 10 mm/s and feed rate of 2 μm/s. As shown in Figure 13, there is no residual recast layer on the machined surface in both the cross-section and longitudinal section of the slit.

### 4.5. Manufacturing of Typical Structure

The typical structure is successfully manufactured by WECDMM, as shown in Figure 14. WECDMM used the NaNO_3_–glycol solution with a conductivity of 30 μS/cm. The other parameters are as follows: voltage of 45 V, pulse on-time of 5 μs, pulse off-time of 12 μs, wire traveling speed of 10 mm/s and feed rate of 2 μm/s. The outline of the pentagram structure is clear. The angles of the four angles of the pentagram structure are 36.12°, 38.44°, 36.62° and 35.61°, respectively. This error may be caused by the movement of the wire electrode caused by the discharge.

## 5. Conclusions

The proposed wire electrochemical discharge micro-machining (WECDMM) removes the recast layer and optimizes surface quality. In this study, the mechanism of WECDMM was elucidated for the first time. The conclusions are as follows:The electrolyte must not only achieve an electrochemical reaction but also meet the requirements of discharge. The electrolyte was optimized through a comparison experiment, and NaNO_3_–glycol solution was determined as the best working solution.Through the observation of machining phenomena and waveform analysis, the discharge characteristics in low conductivity dielectric were investigated. The type of discharge is related to the arrangement of bubbles in the machining gap before discharge. Discharge breakdown of the low conductivity NaNO_3_–glycol solution in an environment of few bubbles. The medium of discharge breakdown is a bubble in an environment of numerous bubbles.The influences of key process parameters including conductivity of the electrolyte, pulse voltage, pulse-on time and wire feed rate were analyzed on the slit width, standard deviation, radius of fillet at the entrance of the slit and roughness.According to the influence on the machining results, the parametric combination is as follows: the NaNO_3_–glycol solution with a conductivity of 30 μS/cm, a voltage of 45 V, pulse on-time of 5 μs, pulse off-time of 12 μs, wire traveling speed of 10 mm/s and feed rate of 2 μm/s. Typical microstructures were machined, which verified the machining ability of WECDMM.

## Figures and Tables

**Figure 1 micromachines-14-01505-f001:**
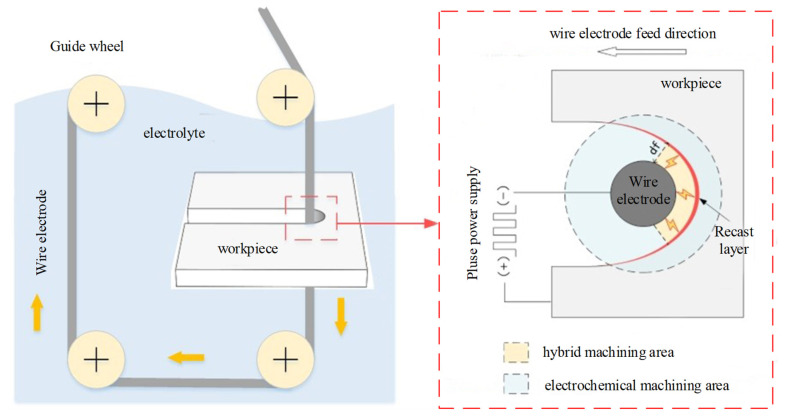
Schematic diagram of WECDMM.

**Figure 2 micromachines-14-01505-f002:**
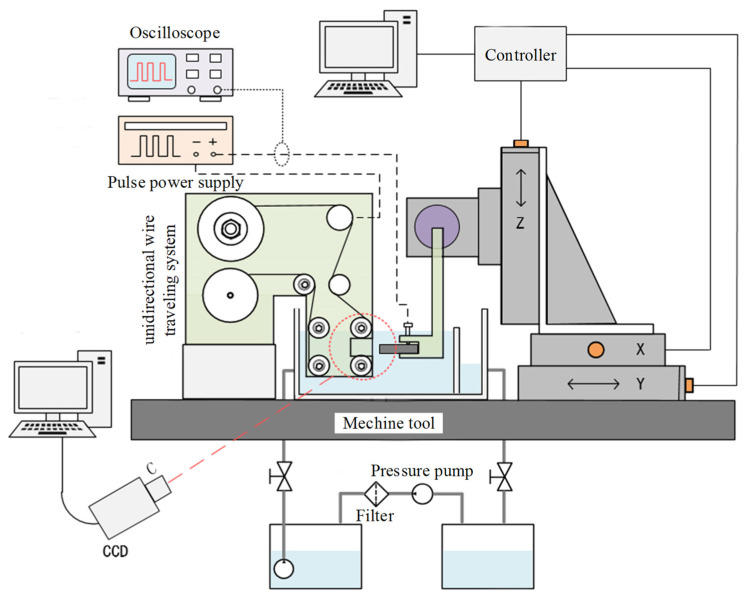
Schematic of experimental system.

**Figure 3 micromachines-14-01505-f003:**
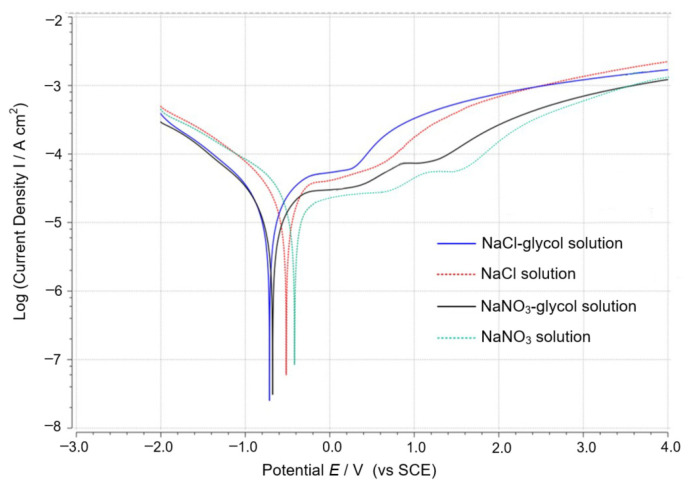
Typical curve of 9Cr18Mo in different electrolytes.

**Figure 4 micromachines-14-01505-f004:**
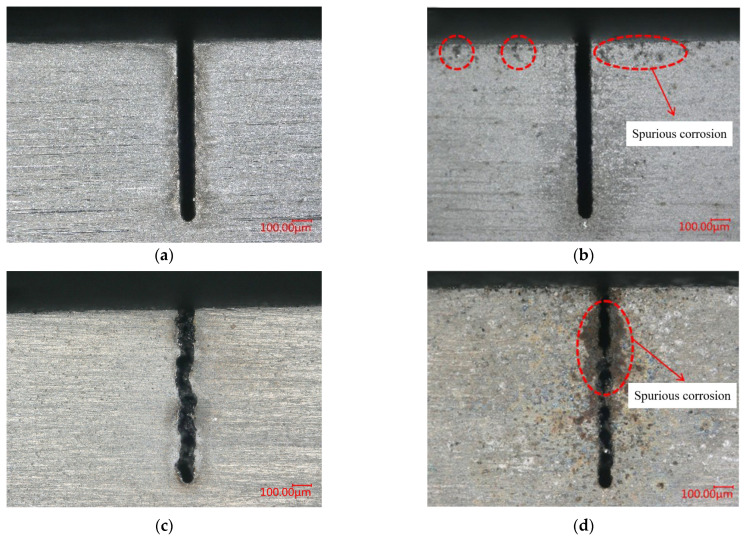
Morphology of micro slits machined by WECDMM with different salt solutions: (**a**) NaNO_3_–glycol solution; (**b**) NaCl–glycol solution; (**c**) NaNO_3_–aqueous solution; (**d**) NaCl–aqueous solution.

**Figure 5 micromachines-14-01505-f005:**
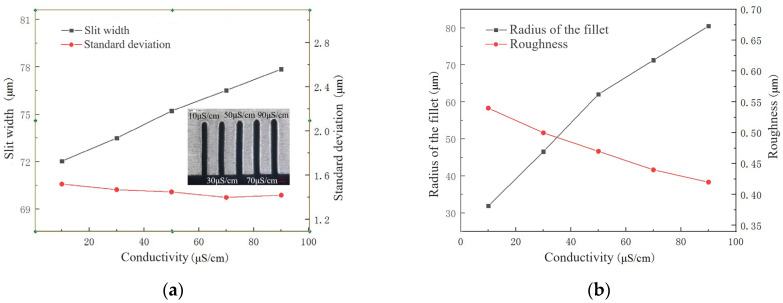
The influence of electrolyte conductivity during WECDMM: (**a**) The influence of electrolyte conductivity on slit width and standard deviation; (**b**) The influence of electrolyte conductivity on roughness and radius of the chamfering.

**Figure 6 micromachines-14-01505-f006:**
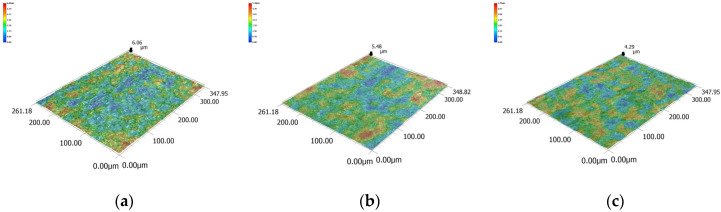
Micro-morphology of slit sidewall machined using NaNO_3_–glycol solution with various conductivities: (**a**) 10 μS/cm; (**b**) 50 μS/cm; (**c**) 90 μS/cm.

**Figure 7 micromachines-14-01505-f007:**
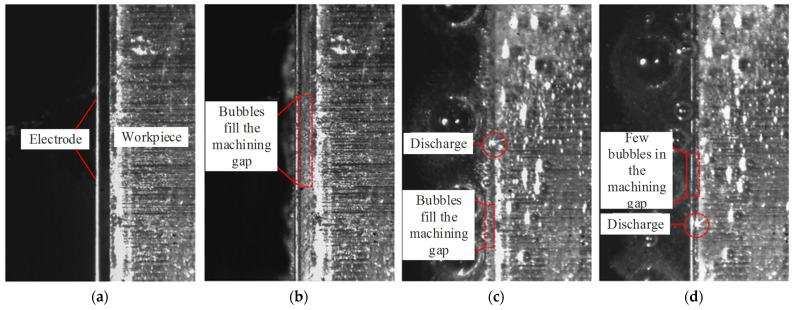
The machining phenomenon of WECDMM: (**a**) Before machining; (**b**) Bubbles generated by electrochemical reaction; (**c**) Discharge occurs in numerous bubbles environment; (**d**) Discharge occurs in few bubbles environment.

**Figure 8 micromachines-14-01505-f008:**
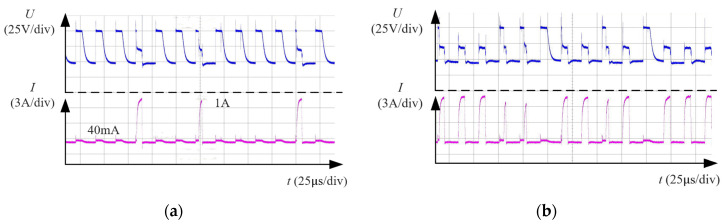
Voltage and current waveform of WECDMM: (**a**) Sparse discharge; (**b**) Dense discharge.

**Figure 9 micromachines-14-01505-f009:**
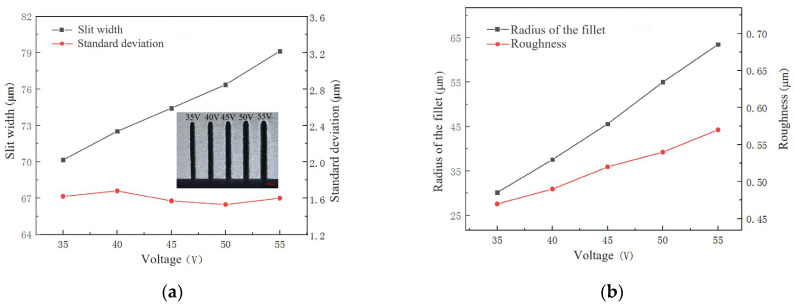
The influence of voltage during microwire electrochemical discharge machining: (**a**) Variation in the slit width and standard deviation; (**b**) Variation in the roughness and radius of the chamfering.

**Figure 10 micromachines-14-01505-f010:**
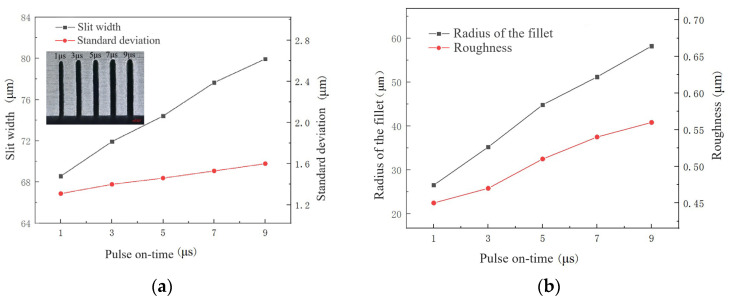
The influence of pulse on-time during WECDMM: (**a**) Variation in the slit width and standard deviation; (**b**) Variation in the roughness and radius of the chamfering.

**Figure 11 micromachines-14-01505-f011:**
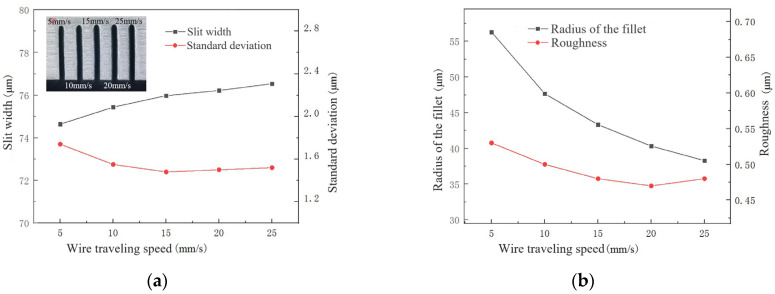
The influence of wire travelling speed during WECDMM: (**a**) Variation in the slit width and standard deviation; (**b**) Variation in the roughness and radius of the chamfering.

**Figure 12 micromachines-14-01505-f012:**
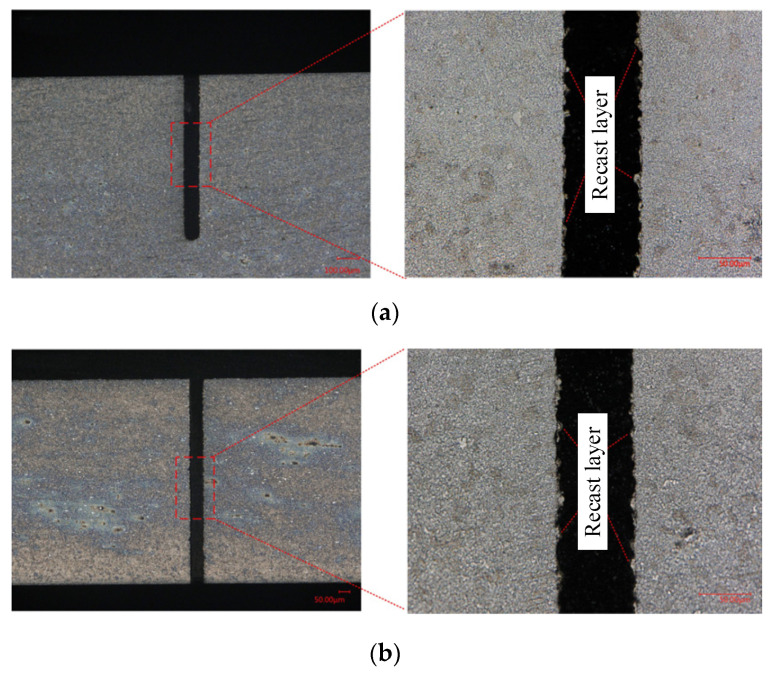
Metallographic microscope images of the slit machined by MWEDM: (**a**) Metallographic microscope images of slit cross-section machined by MWEDM; (**b**) Metallographic microscope images of slit longitudinal section machined by MWEDM.

**Figure 13 micromachines-14-01505-f013:**
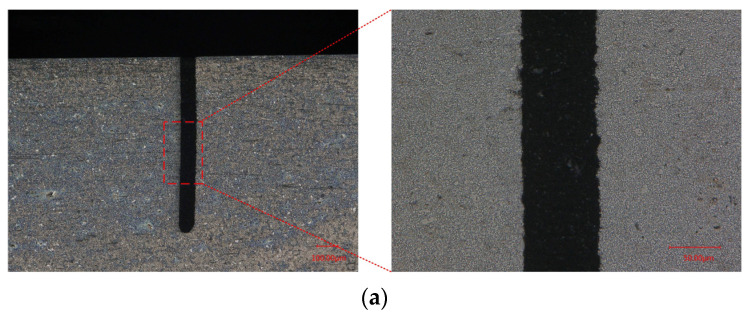
Metallographic microscope images of the slit machined by WECDMM with NaNO_3_–glycol solution with conductivity of 10 μS/cm: (**a**) Metallographic microscope images of slit cross-section machined by WECDMM; (**b**) Metallographic microscope images of slit longitudinal section machined by WECDMM.

**Figure 14 micromachines-14-01505-f014:**
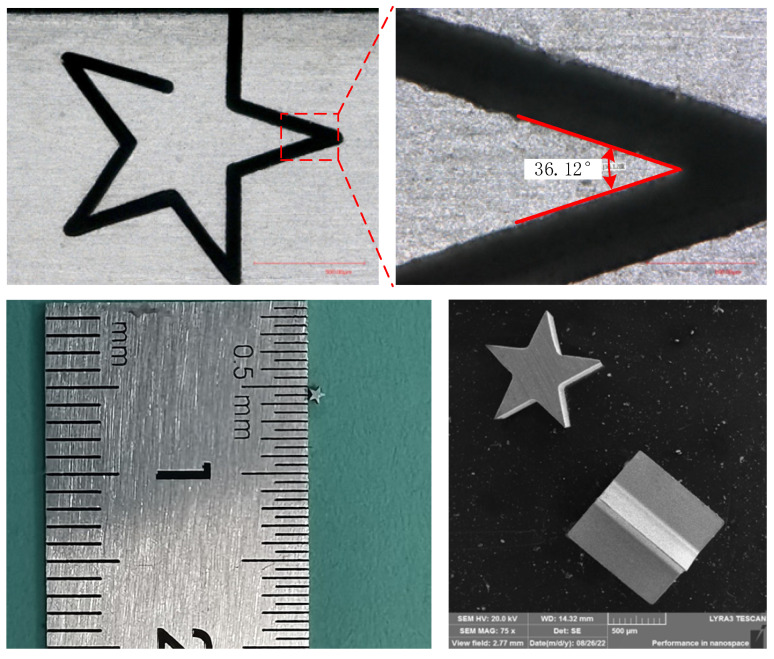
Pentagram structure machined by WECDMM.

**Table 1 micromachines-14-01505-t001:** Experimental parameters.

Item	WECDMM
Work material	9Cr18Mo
Workpiece thickness (mm)	1 mm
electrode (mm)	Φ50 μm tungsten wire
Pulse voltage (V)	35, 40, 45, 50 V, 55 V
Pulse on-time (μs)	1, 3, 5, 7, 9
Pulse off-time (μs)	12
Pulse frequency (kHz)	50
Wire traveling speed (mm/s)	5, 10, 15, 20, 25
Feed rate (μm/s)	2
Conductivity of electrolyte (μS/cm)	10, 30, 50, 70, 90

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
