# Peer review of "Investigation on Wire Electrochemical Discharge Micro-Machining"

_micromachines, 2023, doi:10.3390/mi14081505_

Round 1
Reviewer 1 Report
1.The font size in the title is inconsistent.
2.The content of the title is too broad. It is recommended that the author modify the title to make it more targeted.
3.The header of Table 1 is not centered.
4.Two 4.4 headings appear in the subheading.
5.It is recommended to only indicate figures a, b, c, and d in the figure, and place the explanation of the figure after the title (such as Figure 7).
1. Figure 7. the machining phenomenon of WECDMM. The first letter of “the“” should be capitalized.
Reviewer 2 Report
This manuscript presents quality characterization work on the wire electrochemical discharge micro-machining (WECDMM). This is interesting to micromachines. It may be reconsidered for publication after a minor revision. Detailed comments are as follows:
1. In the fig.5(b), Fig9-11(b), authors defined as “Radius of fillet”, but in the text was decribed as “Radius of chamfering”. The description should be consistent.
2. The servo feed method should be mentioned in the manuscript.
3. The conclusion should be rewritten to reflect important results in addition to general concepts.
Reviewer 3 Report
The author propose the wire electrochemical discharge micro-machining (WECDMM) and develop a electrolyte system, which remove the recast layer. Typical microstructures were machined, which verified the machining ability of WECDMM. Some suggestions and questions are as follows:
1、EDM can remove the workpiece materials, but will it also affect the passivation film on the surface of the workpiece? It is recommended that the author conduct in-depth research on the impact on electrochemical machining.
2、When the machining gap d ≤ df, in hybrid machining area, discharge and electrical reaction currently exists, and discharge is dominant, is there any relevant research to support it?
3、Figure 4 is the morphology of micro slits machined by WECDMM with different salt solutions, but the corresponding description cannot be found in the manuscript.
4、The structure of pentagram is successfully manufactured by WECDMM,the error of angles may be caused by the movement of the wire electrode,so how to control the error?
The author needs to make minor editing to the sentence structure of the article to improve readability.
